🔓 | **Open Peer Review** | Environmental Microbiology | Research Article

# Grass-legume mixtures maintain forage biomass under microbial diversity loss via gathering *Pseudomonas* in root zone soil

Yu Liu,[1] Wei Yan,[2] Tongyao Yang,[1] Yining An,[1] Xiaomeng Li,[1] Hang Gao,[1] Ziheng Peng,[1] Gehong Wei,[1] Shuo Jiao[1]

**ABSTRACT** As the most common approach for the restoration of degraded grasslands, the use of grass-legume mixtures has long been recognized for its role in increasing aboveground biomass and resisting grassland degradation. However, whether the legumes in these mixtures can help neighboring plants resist the decline in biomass caused by the loss of soil microbial diversity remains a question worthy of investigation. To address this, we employed a dilution method to create a gradient of decreasing microbial diversity in soil and utilized full-factorial combinations of legumes and two grasses to investigate the crucial role of legumes in the mixture. The results showed that compared to monoculture, the mixture of *Medicago sativa* L. and *Elymus dahuricus* Turcz. enhanced the biomass of grass species under conditions of soil microbial diversity loss. We then discovered that a significantly enriched *Pseudomonas* (ASV53), in the grass-legume mixtures under conditions of microbial diversity loss, was positively correlated with plant biomass and nitrogen-fixing (*nifH*) gene abundance, implying that it could be a keystone species. In addition, the grass-legume mixture increased the deterministic processes of microbial community enrichment in the root zone soil by enhancing the process of homogeneous selection. Functional predictions revealed that grass-legume mixtures increased the potential abundance of N-related and phototrophy-related microbial communities in the root zone soil. This study provides an important insight into the mechanism underlying the role of legumes in increasing and maintaining grass biomass despite soil microbial diversity loss.

**IMPORTANCE** Grass-legume mixtures are a common practice for establishing artificial grasslands, directly or indirectly contributing to the improvement of yield. In addition, this method helps maintain soil and plant health by reducing the use of chemical fertilizers. The impact of grass-legume mixtures on yield and its underlying microbial mechanisms have been a focus of scientific investigation. However, the benefits of mixtures in the context of soil microbial diversity loss remain a problem worthy of exploration. In this study, we examined different aboveground and belowground diversity combinations to elucidate the mechanisms by which grass-legume mixtures help maintain stable yields in the face of diversity loss. We identified the significantly enriched *Pseudomonas* genus microbial ASV53, which was gathered through homogeneous selection and served as a keystone in the co-occurrence network. ASV53 showed a strong positive correlation with biomass and the abundance of nitrogen-fixing genes. These findings provide a new theoretical foundation for utilizing grass-legume mixtures to enhance grass yields and address the challenges posed by diversity loss.

**KEYWORDS** microbial diversity loss, grass-legume mixtures, root zone, microbial community, nitrogen fixation bacteria

Grasslands are one of the most important biomes on Earth, covering nearly one-third of the planet's land surface (1). They provide a wide range of ecological,

Address correspondence to Gehong Wei, weigehong@nwsuaf.edu.cn, or Shuo Jiao, shuojiao@nwsuaf.edu.cn.

Yu Liu and Wei Yan contributed equally to this article. Author order was determined alphabetically.

The authors declare no conflict of interest.

See the funding table on p. 15.

economic, and social benefits, including carbon sequestration, livestock grazing, and biodiversity conservation (2). However, grasslands are facing numerous threats, with degradation being one of the most significant challenges. Grassland degradation is the loss of grassland productivity and biodiversity due to various factors such as overgrazing, land-use change, climate change, and invasive species (3). This phenomenon has become a global concern, with significant impacts on the environment and human societies; consequently, the restoration of degraded grasslands has emerged as a pressing concern among scientists and local governments (4). Restoration management of degraded grasslands typically involves a range of techniques, including fencing to exclude large herbivores, raking to remove dead vegetation and reseeding with appropriate plants, fertilization to enhance soil fertility and productivity, turf transplantation to establish new vegetation patches, as well as controlling rodent and weed populations to minimize competition for resources (5). Among these practices, the establishment of artificial or semi-artificial grasslands is currently the most widely used method, as it can select suitable grass species, improve grassland productivity, and relieve the pressure on natural grassland (6).

Currently, grass-legume mixtures are the most commonly used planting pattern for artificial grasslands compared with traditional monoculture (7, 8). The key reason for selecting legumes as the main crop for a mixture in agriculture is their ability to fix nitrogen, which is highly valued for enhancing soil fertility and reducing dependence on synthetic nitrogen fertilizers (9). However, not all the nitrogen fixed by legumes necessarily flows to neighboring plants. Studies have found that when the soil environment changes, legumes will prioritize their use of fixed nitrogen, rather than exhibiting the "good neighbor" role as previously believed (10). Although there have been numerous studies exploring the benefits and mechanisms of grass-legume mixtures on grass yield, most of these studies have primarily focused on investigating the impact of different aboveground plant combinations (11, 12). The enrichment of underground microbial communities in response to these combinations and their potential role in helping plants resist the adverse effects of diversity loss has been largely overlooked (13).

Plants and microbial communities have a long co-evolutionary history, and their interactions play crucial roles in shaping soil microbial structures and ecosystem functioning (14). One important way that plants influence soil microbial communities is via the release of root exudates, which can attract or repel different microbial taxa (15). As a result, plants can selectively enrich specific microbial groups in the root zone, leading to a plant-specific microbial community structure. Due to their beneficial effects on plant growth and health, plant growth-promoting bacteria (PGPR) have been extensively studied as potential biofertilizers and biocontrol agents for various crops (16–18). Within the aforementioned PGPR, the most notable are the nitrogen-fixing bacteria that assist plants in acquiring nitrogen from the environment. Apart from the symbiotic nitrogen-fixing bacteria that trigger differentiated structures on the host plant (root nodules of legumes and actinorhizal plants), there is also a diverse range of free-living nitrogen-fixing bacteria that can associate with the root system of graminaceous plants, such as *Klebsiella pneumoniae*, *Azotobacter vinelandii*, and *Azospirillum brasilense* (19). *Pseudomonas stutzeri A1501* can assist with nitrogen fixation in the Poaceae family, further expanding the range of nitrogen-fixing bacteria. *P. stutzeri A1501* can colonize the root surface and invade the superficial layers of the root cortex (20). However, it is not clear whether the root zone of grasses will become enriched with specific microbial communities due to the influence of adjacent legumes within mixed grass-legume plantings. Nonetheless, plant species and cultivation methods do affect the community structure of root zone microbes, which has theoretical guidance significance in agricultural production (21), as such, it is important to clarify the role of legumes in these mixed plantings.

Members of Poaceae and Fabaceae are the representative species types in grasslands, with *Medicago sativa* L. (legume), *Elymus dahuricus* Turcz. (grass), and *Festuca elata* Keng. (grass) being three commonly used forage species for the establishment of artificial

pastures (22–24). In this study, we used these three plants for pot cultivation under different plant combinations and soil microbial diversities. The use of pot experiments enables the identification of microbial communities that have a significant impact on plant growth, as they provide a more sensitive reflection of microbial changes within the root zone (25). We systematically investigated the enriched microbial communities of the root zone soil in the grass-legume mixture. Our first objective was to determine whether the grass-legume mixture could help enhance the yield of grass in the context of soil microbial diversity loss. Subsequently, we intended to decipher the underlying mechanisms behind this phenomenon by focusing on the significantly enriched microbial taxa. Our findings revealed that the mixture exhibited deterministic enrichment of the *Pseudomonas* genus member *ASV53* in the face of diversity loss. This organism showed a significant positive correlation with biomass and N-related functions. Through the assistance of these communities, the grass was able to mitigate the negative impacts of soil microbial diversity loss and maintain biomass stability.

## RESULT

### The construction of a gradient for soil microbial diversity and the impact of grass-legume mixtures on biomass

We successfully created a loss gradient of soil microbial diversity and found that the microbial α-diversity in the root zone decreased with increasing dilution (Fig. S1a), and significant differences in β-diversity were observed across soil dilution gradients (Fig. S1b). When comparing the biomass of grass under monoculture and grass-legume mixtures conditions, we found that grass-legume mixtures (MP) resulted in a significant ($P < 0.05$) increase in the total biomass compared to monoculture (P) (Fig. 1a). To further ascertain the factors contributing to the total biomass increase, we compared the individual plant weights of grasses and legumes under monoculture and in mixture. The results revealed that the increase in individual plant biomass was more pronounced in the grass than in legumes (Fig. 1b). This was confirmed even under conditions of soil microbial diversity loss, as the individual weights of both grass (M) and legume (P) decreased with the decline in microbial diversity in monoculture, the grass-legume mixtures (MP) increased the biomass of grass and maintained its biomass stability despite the reduction in soil microbial diversity (Fig. 1c). These findings suggest that grass-legume mixtures can be an effective approach for improving productivity in the establishment of artificial grasslands. In addition, due to the involvement of legumes, we also compared the content of ammonium and nitrate nitrogen in the soil under different plant combinations and found no significant difference ($P > 0.05$) between monocultures and the mixtures (Fig. S3). This suggests that the increase in grass biomass under the mixture was not achieved by changing the physicochemical properties of the soil.

### Microbial community diversity, structure, and composition of grass-legume mixtures

To investigate the impact of different plant combinations on the composition and structure of root zone soil microbial communities, we first compared the differences in microbial α-diversity between monoculture and grass-legume mixtures. The observed richness, Shannon, and inverse Simpson indices showed no significant difference ($P > 0.05$) between monocultures and grass-legume mixtures, indicating that microbial diversity may not be the main driver of yield benefits from the mixture (Table S2). We then used the significance test methods of multi-response permutation procedure (MRPP), analysis of similarities (ANOSIM), and permutational multivariate analysis of variance (PERMANOVA) to explore the β-diversity of root zone soil microbial communities under different plant combinations based on Bray-Curtis distance (Table S3). The results showed that the microbial communities in monocultures and the mixtures were not significantly different ($P > 0.05$) (Fig. S4).

To summarize, there were no significant differences in microbial α- and β- diversity of the root zone between monocultures and grass-legume mixtures. However, there was a

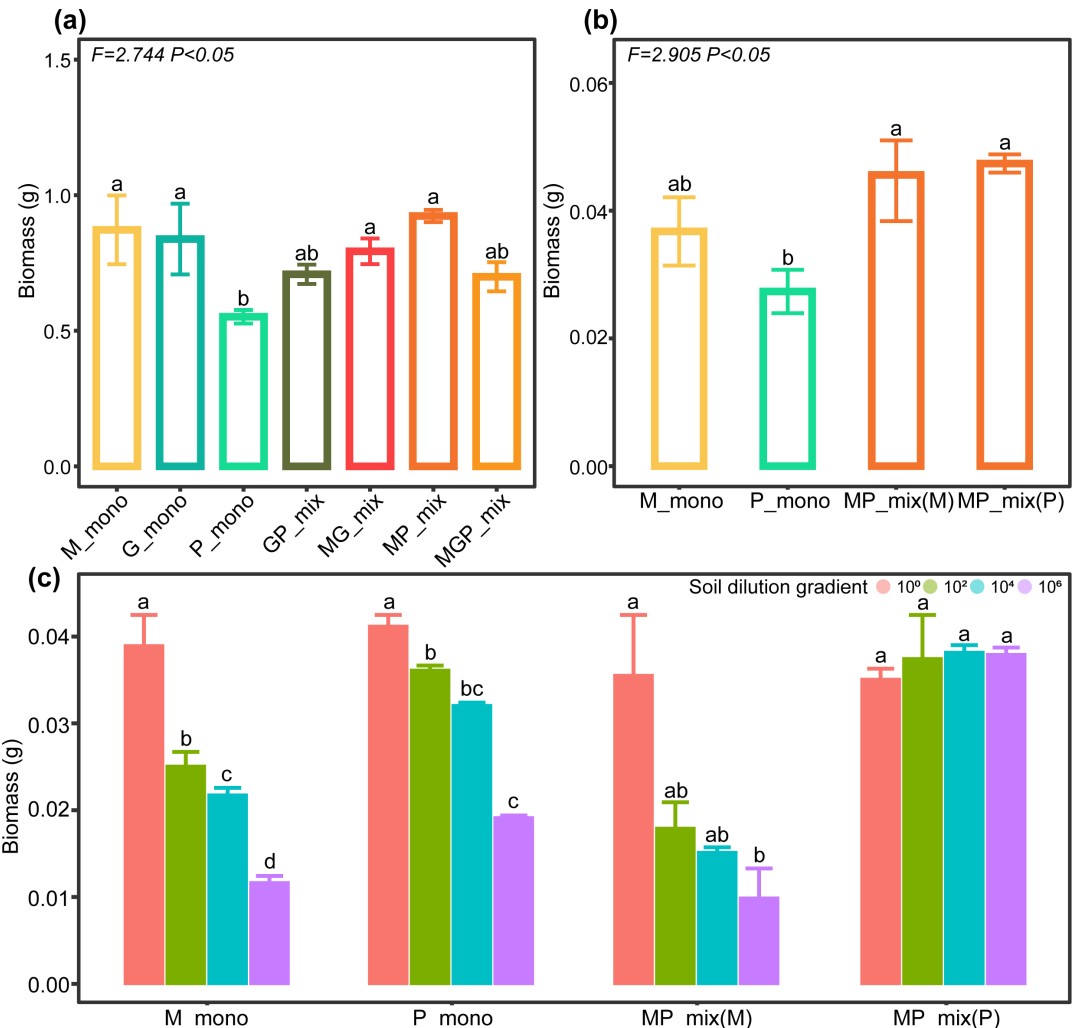

FIG 1 The variation in biomass under different plant combinations. Lowercase letters indicate significant differences. M_mono, *Medicago sativa* L. monoculture; G_mono, *Festuca elata* Keng. Monoculture; P_mono, *Elymus dahuricus* Turcz. monoculture; GP_mix, *Festuca elata* Keng. and *Elymus dahuricus* Turcz. mixture; MG_mix, *Medicago sativa* L. and *Festuca elata* Keng. mixture; MP_mix, *Medicago sativa* L. and *Elymus dahuricus* Turcz. mixture; MGP_mix, a mixture of *Medicago sativa* L., *Festuca elata* Keng. and *Elymus dahuricus* Turcz.. (a) The overall changes in biomass within the seven plant cultivation patterns. (b) Comparison of the individual plant weight of *Medicago sativa* L. and *Elymus dahuricus* Turcz. when grown separately versus in a mixture. Analyses of variances (ANOVAs) were conducted to test for significant differences, with *F* values and their significance given in the top left-hand corner (*P* < 0.05). (c) The changes in individual plant weight of *Medicago sativa* L. and *Elymus dahuricus* Turcz. under the condition of soil microbiome diversity loss. The values in the figure legend named $10^0$, $10^2$, $10^4$, and $10^6$, represent the dilution gradients of the soil suspension. Specifically, $10^0$ signifies the undiluted original mother solution, $10^2$ corresponds to a 100-fold dilution, $10^4$ denotes a 10,000-fold dilution, and $10^6$ signifies a 100,000-fold dilution.

significant increase in biomass for the grasses in the mixture; thus, we believe that there must be some soil microbial community members that play a crucial role despite their insignificant diversity changes. Therefore, a classifier program was used to annotate the root zone soil microbial communities, and comparisons were made at the phylum and genus levels under different plant combinations. All ASVs were classified into 525 genera and 42 phyla. We found that the microbial composition in the root zone of different plant combinations was relatively similar. Proteobacteria (59.75%) were dominant (Fig. S2a) in P monoculture compared with the MP mixture. The *t*-test results showed that Bacteroidota were significantly less abundant in the mixtures (11.28%) than in monocultures (16.35%) (Fig. S2b). At the genus level, *Pseudomonas* (23.28%) was dominant among different plant combinations (Fig. S2c), and we found the abundance of *Pseudomonas* was significantly higher in MP mixture (26.35%) than in P monoculture (14.38%) (Fig.

S2d). *Acinetobacter*, a genus known to contain several pathogens, was ubiquitous in the root zone soils and its relative abundance in MP mixture was 0.46%, while in P monoculture it was 4.13% (Fig. S2d). To further determine the role of *Pseudomonas* in the mixture plantings, we also used the method of linear discriminant analysis effect size (LEfSe) to identify microbial taxa that showed significant changes across different soil dilution gradients. The results indicated significant enrichment of *Pseudomonas* taxa by plants at a dilution of $10^6$ (Fig. 2a). Finally, we utilized the tree map to integrate and present the results of both direct comparison of community relative abundance and LEfSe analysis results (Fig. 2b). The above results implied that the grass-legume mixtures increased the relative abundance of potential PGPR, such as *Pseudomonas*, and reduced the relative abundance of pathogenic bacteria (e.g., *Acinetobacter*).

## The enriched *Pseudomonas* genus member ASV53 in the grass-legume mixtures and its functions

To ascertain which specific bacteria within the *Pseudomonas* genus played an important role, we conducted a differential analysis comparing the ASVs that showed significant changes among M and P monocultures and the MP mixture. Our analysis showed that *ASV53* was remarkably enriched in the mixture (Fig. 3a). We then conducted the correlation between the abundance of *ASV53*, the copy number of *nifH* genes, and the individual plant biomass of grass under the MP mixture. The results revealed a significant positive correlation between the abundance of *ASV53* and the copy number of *nifH* genes ($R^2 = 0.9764$, $P < 0.05$), as well as the biomass ($R^2 = 0.9543$, $P < 0.05$) of grass (Fig. 3b). A previous study showed that *P. stutzeri A1501* could help fix nitrogen in the Poaceae family, expanding the understanding of nitrogen-fixing microbes (20). Given the strong correlation between *ASV53* and nitrogen fixation genes in our study, we constructed a phylogenetic tree of all annotated *Pseudomonas* sequences and the downloaded 16S rRNA gene sequences of the *A1501* strain from NCBI and found that *ASV53* and *A1501*

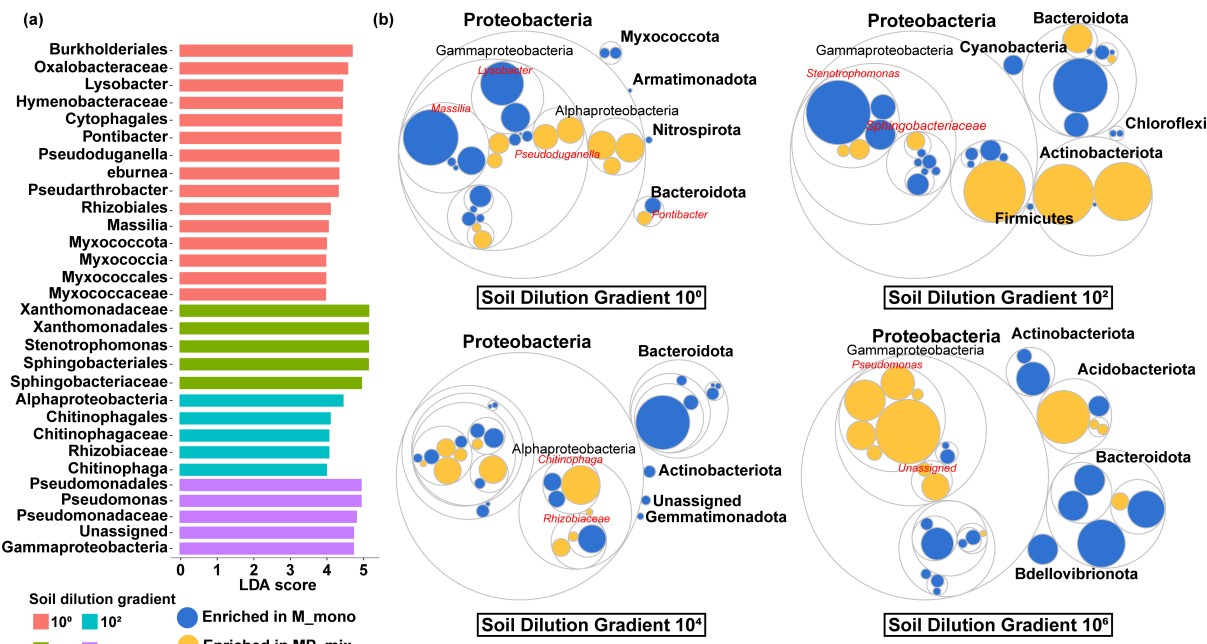

**FIG 2** We compared microbial taxa that exhibited significant changes across soil dilution gradients and identified which taxa were enriched in monoculture and mixtures. (a) LEfSe plots showing bacterial abundance enriched in monoculture and mixtures. Histograms of different colors indicate taxa (ranging from the genus level to the class level) that were abundant in the corresponding soil dilution gradient ($P < 0.05$); (b) The circular packing chart of taxonomic differences between monoculture and mixture soils under different soil dilution gradients. The outermost to innermost circles represent phylum, class, order, and family levels, respectively. The red font lists the genera that were the focus of this study. Solid circles represent genera. Bacterial taxa that were significantly enriched are indicated in dark blue for monoculture and yellow for mixtures.

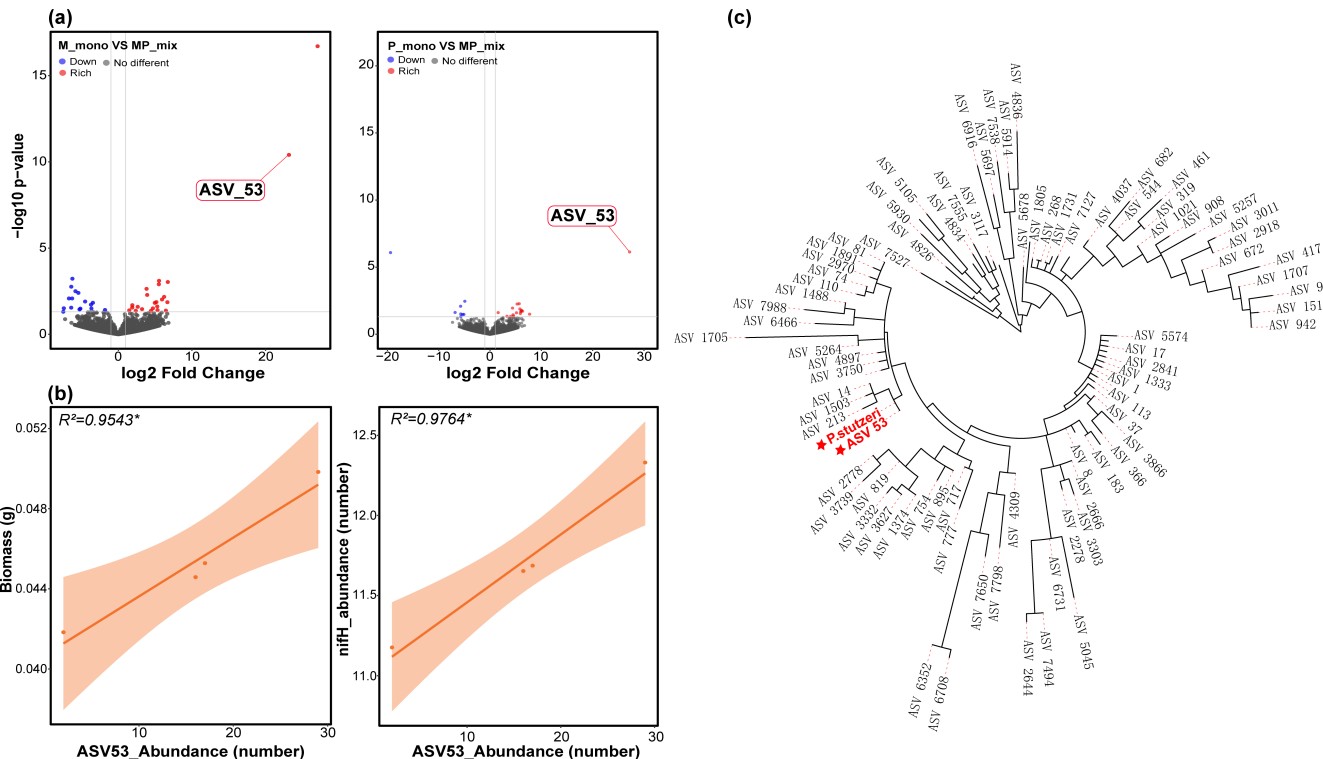

**FIG 3** Identifying the key microbes under different plant combinations and exploring their functions. (a) ASVs exhibiting significant differential abundance were identified through differential analysis among monocultures of M_mono and P_mono, and the MP_mix. (b) Investigation of the relationship between the abundance of ASV53 and the individual plant biomass of *Elymus dahuricus* Turcz., as well as the abundance of the nitrogen fixation gene *nifH* in the nitrogen cycling. (c) Construction of an evolutionary tree to determine the phylogenetic relationship between all obtained *Pseudomonas* ASVs in our study and *P. stutzeri A1501*, a known nitrogen-fixing *Pseudomonas* species.

had a very close relationship (Fig. 3c). As the sequence information for *ASV53* is V4-V5 region derived from high-throughput 16S rRNA sequencing data, we selected a segment of the 16S rRNA from the *A1501* whole-genome sequence and aligned it with *ASV53*. The similarity between *ASV53* and *A1501* was 98.12%. The Pearson correlation heatmap also indicated that *ASV53* had a strong correlation with the *nifH* gene in all plant combinations, but only in the MP mixture did *ASV53* show a significant positive correlation with plant biomass (Fig. S5a through c).

We then delved into the role of *ASV53* in the entire microbial co-occurrence network. In general, the MP mixture increased the number of nodes and edges in the network, resulting in more positive correlations between nodes and promoting network stability. In addition, different plant combinations significantly altered the modules within the co-occurrence network (Fig. 4). More specifically, the MP mixture transformed the network's dominance from modules 1 and 5 to module 4, and *Pseudomonas ASV53* became a crucial key node in the network, with the highest degree (Fig. 4a). To examine the role of *ASV53* further, we introduced functional gene data into the network and scrutinized the correlation between nitrogen cycling-related genes and network nodes in the context of the MP mixture. The outcomes revealed a significant positive relationship between *ASV53* and the nitrogen fixation gene *nifH* (Fig. S6b). Furthermore, while assessing the changes in nitrogen cycling-related genes in the root zone soil across various plant combinations, we observed that the MP mixture considerably boosted the soil's nitrogen fixation gene *nifH* content (Fig. S5). All the results mentioned above indicated that the MP mixture can help enrich the *Pseudomonas ASV53* in the root zone soil of grass and that it has nitrogen-fixing ability, contributing to the biomass of grass.

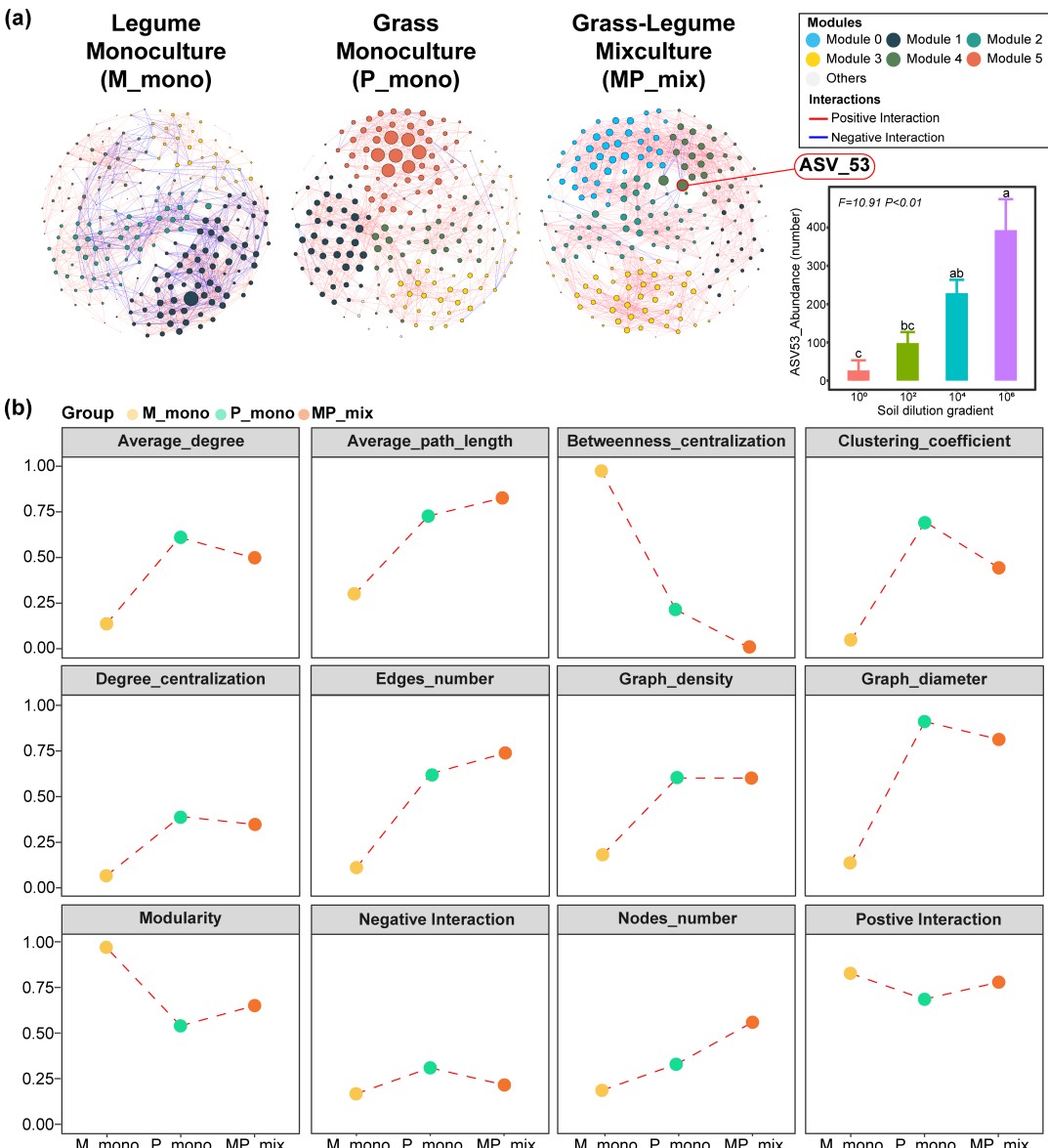

**FIG 4** The changes in microbial co-occurrence networks and the role of key microbe ASV53 within different plant combinations. (a) The variations in co-occurrence networks between monocultures of M and P, and MP mixture, as well as the changes in the abundance of ASV53 under the condition of soil microbiome diversity loss. (b) The changes in the basic properties of the network structure in three plant combinations.

## Bacterial community assembly process under monoculture and grass-legume mixtures

We used null models to investigate whether the mixtures could bring about changes in community assembly processes and further confirmed the effects of the mixture on the root zone soil microbial communities. Undominated processes contributed the largest fraction to the assembly of all plant combinations, in our focus on M and P monocultures and their mixture, the proportions were 61.05%, 57.78%, and 53.03%, respectively (Fig. 5). Furthermore, the MP mixture resulted in a more deterministic community assembly process (Homogeneous Selection) of root zone soil microbes, increasing from 20.53%, 13.33% under monoculture to 33.33%, indicating the enrichment of specific bacteria by plants. Furthermore, we also predicted the potential microbial functions in the root zone soil under M and P monocultures and MP mixture. The results were consistent with the previous results, showing enrichment of nitrogen-fixing-related functions, and

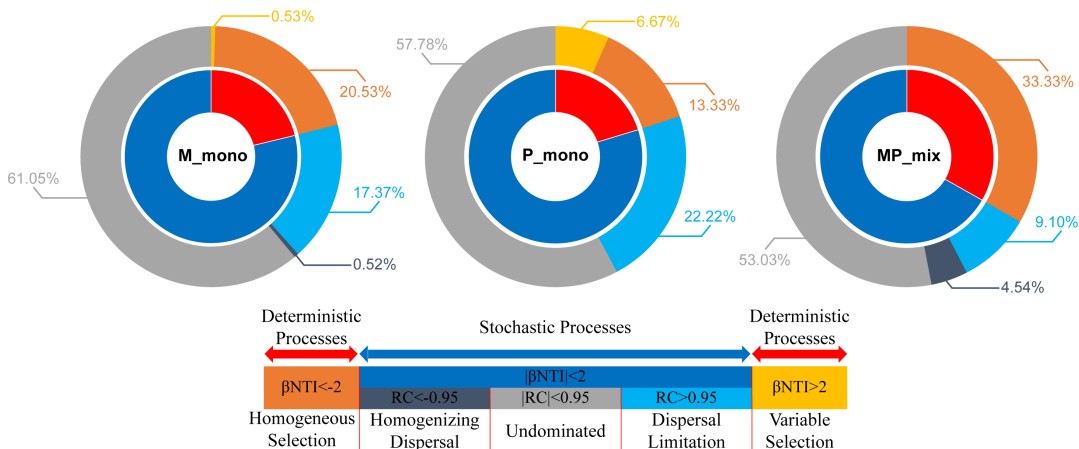

**FIG 5** Assembly of the monocultures of M_mono, P_mono, and MP_mix. The inner circle represents the contribution of stochastic and deterministic processes to bacterial assembly. The outer circle represents the percentage of detailed ecological processes belonging to stochastic or deterministic processes.

a decrease in potential disease risk (Fig. 6). Based on all the results above, we concluded that the MP mixture could help to more deterministically enrich microbial communities with nitrogen-fixing functions in root zone soil, particularly for grass (Fig. 7).

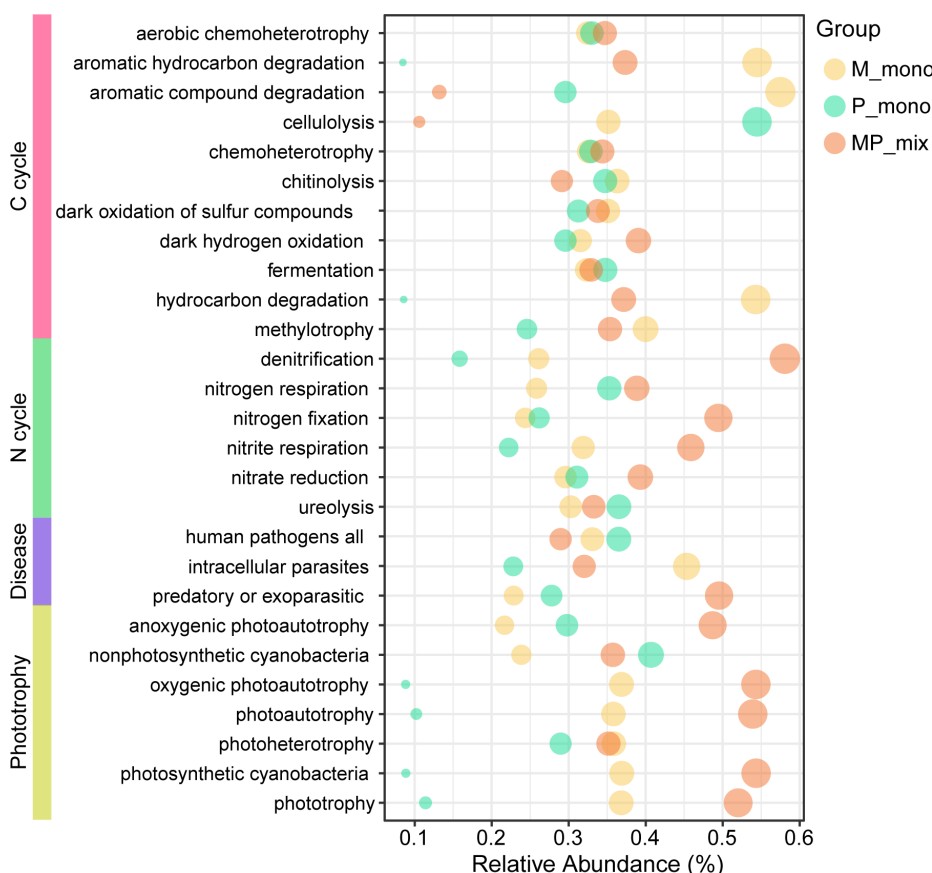

**FIG 6** Prediction of soil microbial functions under the scenarios of monocultures of M and P, and MP mixture. The respective functionalities, from top to bottom, are as follows: carbon cycling-related functions, nitrogen cycling-related functions, disease-related functions, and light-related functions.

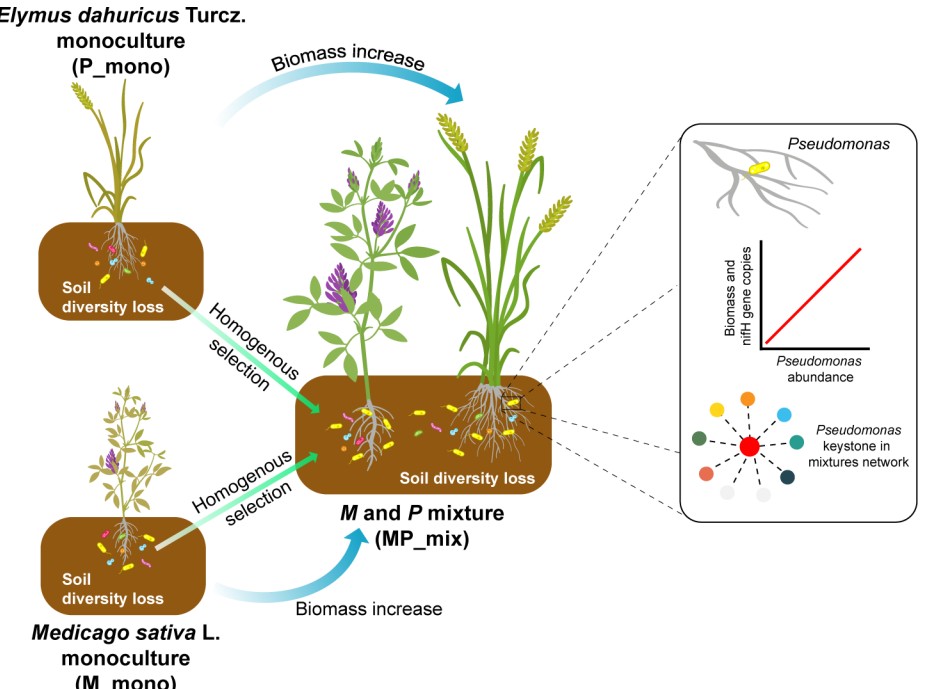

**FIG 7** The model diagram depicts the phenomenon of grass-legume mixture aiding in biomass enhancement and enriching *Pseudomonas* of root zone soil under conditions of microbial diversity loss. When *Elymus dahuricus* Turcz. (P) and *Medicago sativa* L. (M) were grown individually, both species experienced a decrease in biomass under conditions of soil microbial diversity loss. However, in the grass-legume mixture, homogeneous selection led to a significant enrichment of *Pseudomonas* in the root zone. These *Pseudomonas* exhibited a strong positive correlation with biomass and the nitrogen-fixing gene *nifH*, served as the keystone in the microbial co-occurrence network, and contributed to biomass increase.

## DISCUSSION

In this study, we first compared the biomass trends of grass under different plant combinations and found that the MP mixture significantly increased the biomass of grass, and could maintain its production stability under our simulated soil microbial diversity loss. However, this increase in yield was not significantly correlated with soil physicochemical properties, so, we then focused on the diversity and structural differences of root zone microbial communities among the different plant combinations. The results showed no significant differences in α- and β-diversity between monocultures and mixture, but we found that the MP mixture significantly enriched *Pseudomonas* under soil diversity loss. We then identified *ASV53* which had a very close phylogenetic relationship with the previously reported nitrogen-fixing *P. stutzeri A1501*. *ASV53* not only showed an important role in the microbial co-occurrence network but also had a positive correlation with biomass and nitrogen fixation gene abundance in the MP mixture. Moreover, the root zone soil of grass enriched *ASV53* through homogeneous selection, and functional prediction results showed that the mixture soil exhibited a stronger N-cycling potential while reducing the risk of pathogenic bacteria occurrence.

### The yield increase from the grass-legume mixture

The use of grass-legume mixtures is a common practice in both agricultural production and the restoration of degraded grasslands (11, 26). Besides the nitrogen fixation ability of legumes that can increase nitrogen input to neighboring plants, the mechanism underlying the yield increase in grass species through mixture has long been a focus of attention for ecologists and agricultural experts (27). Combining previous long-term field experiments (28) with the conclusions of our study, we found that there were no

significant changes in soil nitrate nitrogen, ammonium nitrogen content, total nitrogen, total carbon, moisture content, and soil conductivity among different plant combinations. This indicates that the grass-legume mixture may not have a direct impact on the physical and chemical properties of soil, and the main reason for the increase in biomass of grass species under mixed planting is not related to soil properties. This could be because the nitrogen fixed by legumes is directed to meet their own needs before being provided to surrounding plants, as confirmed by related research in tropical forests (29). The "good neighbor" role of legumes requires a certain level of soil nitrogen content, and only under conditions of high soil available nitrogen, can legumes reduce competition with neighboring plants, ultimately achieving mutualism and harmonious coexistence (10).

In addition, the loss of soil microbial diversity is also a significant problem faced by grassland degradation (30). Many studies have shown that there is an important relationship between microbial diversity and productivity in the soil (31, 32). Different types of microbial communities play distinct roles, some can decompose organic matter and release nutrients (33), while others may convert these nutrients into forms that can easily be taken up by plants (34). Thus, a more diverse microbial community may promote better nutrient cycling and increase plant nutrient use efficiency, ultimately leading to higher crop yields. Some fungi can form symbiotic relationships with plants, such as mycorrhizal fungi that associate with plant roots and help them absorb nutrients (35, 36). We first compared the variations in biomass across different plant combinations. We observed that the M and P mixture increased the biomass of P, while mixtures had no significant effect on the biomass of G. Similar observations were made with plant combinations GP and MGP. This is the reason why we focused our subsequent analysis exclusively on the individual growth of M and P and their mixed cultivation as MP. The occurrence of these patterns is not only attributed to the crucial role of the soil microbial community, which we have been investigating but also to the inherent characteristics of the plants themselves (37). As commonly used grass species in establishing artificial grasslands, G and P exhibit preferences for specific soil environments and tolerances. Empirical knowledge from livestock farming indicates that P thrives in nutrient-rich soils (38). When P is mixed with M, the nitrogen-fixing capacity of M provides a more abundant nitrogen source, creating a nutrient-rich environment that favors the growth of P. On the contrary, G responds sensitively to soil fertilization, and excessively high nitrogen content in the soil can be detrimental to its growth (39). Thus, a mixture with M does not increase G biomass. In addition, P demonstrates strong adaptability to temperature, while G prefers cooler environments (40). The controlled temperature conditions in our study may not have reached the optimal growth temperature for G. Consequently, there were no significant differences in biomass, whether in monoculture or mixture. In our artificially simulated soil microbial diversity loss experiment, both M and P monocultures showed a decrease in biomass. However, the MP mixtures helped to maintain the stable biomass of the grass, indicating that the mixtures must have formed a protective mechanism to help the root zone microbes resist the adverse effects of microbial diversity loss.

## Grass-legume mixtures changed the composition of the root zone bacterial community from grass

In the restoration processes of degraded grasslands, the selection of plant species and planting methods can vary depending on local soil properties or climatic conditions, and both can be important factors influencing soil microbial communities (41). In our study, after a 2-month pot experiment, we found no significant differences in the α- and β-diversity of soil microbial communities among different plant combinations. This phenomenon could be attributed to multiple factors. First, functional redundancy among soil microbial communities (42) can result in a similar overall functional diversity regardless of changes in plant combinations. Second, the experimental scale, such as pot experiments, may capture changes in microbial taxa with more sensitivity, but the

short duration of the experiment may not have reached the threshold to significantly impact overall α- and β-diversity. Lastly, other environmental factors, including soil type, moisture, temperature, and nutrient availability, apart from plant combinations, can also influence soil microbial communities and their diversity. If these environmental factors are similar across different plant combinations, they may mask any potential differences in the α- and β-diversity of soil microbial communities.

Bacteria are key components of important functions in soil and can have beneficial effects on plant growth through direct or indirect interactions (43). Among them, genera such as *Pseudomonas*, which are known to promote plant growth, can help plants through mechanisms such as nodulation (44), antagonism (45), production of IAA (indole-3-acetic acid) (46), and siderophore (47). We found that the MP mixture significantly increased the abundance of *Pseudomonas* in the root zone soil. In addition, the abundance of *Acinetobacter*, a group of potential pathogens found in a variety of environments, including fresh and marine water, soil, and marine sediments (48), was decreased in the mixture, compared to P monoculture. Altogether, the significant changes in abundance of these two key microbial genera resulted in the increased yield of the grass-legume mixture.

After identifying significant changes at the genus level, we aimed to further investigate which ASV played a dominant role in the process of the MP mixture. We employed microbial co-occurrence network analysis to identify key nodes in different plant combinations and found that *ASV53* had the most connections with other nodes in the network for this mixture. As the *ASV53* sequence was annotated as belonging to the *Pseudomonas* genus in the database, we constructed a phylogenetic tree using the 16S rRNA sequences of all *Pseudomonas* ASVs in our study and the 16S rRNA sequence of *P. stutzeri A1501*, a nitrogen-fixing *Pseudomonas* strain reported in the literature (20). The results showed that *ASV53* was closely related to *A1501*. *P. stutzeri A1501* is a nitrogen-fixing bacterium isolated from paddy fields, which exhibits significant nitrogen-fixing activity under microaerobic conditions, and its nitrogen fixation products can be rapidly absorbed and utilized by rice plants (49). To investigate whether *ASV53* possessed nitrogen fixation and plant growth-promoting abilities similar to *A1501*, we incorporated nitrogen cycling-related functional gene data into the network analysis and found a strong positive correlation between *ASV53* and the nitrogenase gene (*nifH*). Moreover, we found that only in the MP mixture was the abundance of *ASV53* significantly positively correlated with the copy number of *nifH* and biomass. We also found that homogeneous selection was an important process through which grass-enriched *ASV53* in the root zone soil. Although our study did not investigate the specific mechanisms of *ASV53*'s interactions with plants in depth, the analysis results suggest that *ASV53*, similar to *A1501*, plays an important role in nitrogen fixation, helping grasses and leading to increased biomass of grasses and resistance to the adverse effects of decreased biodiversity during mixture planting patterns. This finding could be further validated in subsequent experiments by isolating *ASV53* from the soil of grass-legume mixtures and investigating its genetic and phenotypic characteristics.

## Conclusion

In conclusion, there was a significant increase in the biomass of grass under grass-legume mixtures, and the legume maintained biomass stability when microbial diversity in the soil was reduced. However, grass-legume mixtures showed no significant impact on soil physicochemical properties. To elucidate the underlying mechanism of intercropping on grasses, we conducted differential analysis and network construction and identified that the pseudomonad *ASV53* showed a significant increase in the mixture over the monoculture. We found that *ASV53* was closely related to the nitrogen-fixing *P. stutzeri A1501* and its abundance was significantly correlated with nitrogen-fixing gene copy number and grass biomass under a mixture planting pattern, and it was enriched in grass root zone soil by homogeneous selection. This study reveals new support for the

restoration of artificially degraded grasslands, providing a new perspective on the soil microbial interactions in the root zone during grass-legume mixtures.

## MATERIALS AND METHODS

### Building of model soil

Due to the physical properties of the local farmland soils, soil compaction often occurs in pot experiments. To avoid compaction and repair the physical structure of the soil, making the gaps between soil particles looser for the flow of water and nutrients, we created a model soil by mixing fresh farmland soil (108°09′E, 34°31′N) passed through a 2-mm sieve with an equal amount of fine sand. The model soil was then placed in a double-layered sealed bag, leaving an air outlet. The soil samples in the bag were sterilized in a sterilization pot at 121°C for 90 min and immediately sealed tightly with the hope to prevent airborne bacteria from entering.

### Construction of soil microbial and plant community diversity

The gradient of the diversity of soil microbial communities was constructed using the dilution gradient method of bacterial suspension. After collecting and sieving the soil samples, 100 g of the sample was weighed and placed in 1 L of sterile water, then stirred to fully disperse the soil and obtain a mother liquor with a concentration of 100, which was then sealed and reserved. The mother liquor was then diluted separately with sterile water to achieve dilution factors of $10^2$, $10^4$, and $10^6$. An amount of 300 mL of the different bacterial suspensions was added to each bag of the sterilized model soil, resulting in different gradients of soil microbial diversity (50). To achieve plant diversity, we conducted treatments in each pot with a single plant species, two plant species mixtures, and a three-plant species mixture (51). Each pot contained a total of 24 plants, with 24 plants of each single species under monoculture (5 pots per soil microbial diversity gradient), and 12 plants of each species under two plant mixtures (4 pots per soil microbial diversity gradient), for pots with the three plant species mixture, there were 8 plants of each species (4 pots per soil microbial diversity gradient). For the sake of simplicity, we will use the abbreviations M, P, and G to represent *Medicago sativa* L. (legume), *Elymus dahuricus* Turcz. (grass), and *Festuca elata* Keng. (grass), respectively, in the following text.

### Establishment of a sterilized environment

The planting pots (21.5 cm in diameter and 14.8 cm in height) were covered with a plastic bag with a height of 30 cm, and the interface between the plastic bag and the pot was taped. Two holes were cut on each side of the plastic bag, and a 16 × 16 cm sealing film was attached to the holes with tape to allow for air circulation. First, a layer of non-woven fabric with a diameter of 20 cm was laid in the pot, followed by a 1 cm thick layer of quartz sand on top of the fabric, and another layer of non-woven fabric on top of the sand. After that, the different gradients of microbial diversity soil were added. To avoid contamination from external sources, we used sterilized forceps and gently planted the seeds that had reached the planting conditions into the soil, with a sowing depth of about 2 cm. To ensure consistency and minimize external factors, we maintained a temperature of 25°C, relative humidity of 60% relative humidity (RH), $CO_2$ concentration of 400 PPM, and 12 h of light each day throughout the experiment. We weighed the pot as a whole to determine the amount of water needed and a sterile syringe to spray sterilized water from the top of the bag and evenly watered the plants every 48–72 h. After watering, the small hole was sealed with tape. All equipment used above had been pre-sterilized, and all operations were carried out in a laminar flow hood.

## Sampling and measurement

The root zone soil samples and plant samples were collected after 2 months of planting. A complete root system was dug out to collect root zone soil. Aboveground structures were harvested from each pot, then dried at 70°C for 72 h, and their dry weight was recorded after cooling. Our experiment included different gradients of soil microbial diversity and combinations of different plant species. Under four soil microbial diversity gradients ($10^0$, $10^{-2}$, $10^{-4}$, and $10^{-6}$), seven plant combinations (M, G, P, GP, MG, MP, and MGP) were set, each combination had four replicates. In total, we collected a total of 112 root zone soil samples.

## DNA extraction and amplicon sequencing

DNA was extracted from root zone soil using the FastDNA SPIN Kit for Soil (MP Biochemicals, Solon, OH, USA), following the manufacturer's instructions, and stored at −80°C until use. We performed 16S rDNA amplicon sequencing of the root zone soil samples. For prokaryotic amplicon preparation, the V4-V5 region of the 16S rDNA gene was amplified by PCR using 515F (5′-GTGCCAGCMGCCGCGGTAA-3′) and 907R (5′-CCGTCAATTCCTTTGA GTTT-3′) primers. PCRs were performed in triplicate in a 15-µL reaction mixture which contained 7.5 µL of Phusion High-Fidelity PCR Master Mix (New England Biolabs, Ipswich, MA, USA), 1 µL of forward and reverse primers (3 mM), 2.5 µL of template DNA (5 ng µL−1), and 4 µL of ddH2O. PCR conditions were as follows: 1 cycle × 98°C for 1 min; 30 cycles × 98°C for 10 s, 50°C for 30 s, and 72°C for 30 s; 1 cycle ×72°C for 5 min. The quality of amplicons was detected through electrophoresis on 2% agarose gel, and the purity of amplicons was ensured using the Qiagen Gel Extraction Kit (Qiagen, Hilden, Germany). The triplicate PCRs for each sample were combined and quantified on a QuantiFluor-ST Fluorometer (Promega, Madison, WI, USA) following the manufacturer's protocol. 16S rDNA sequencing was performed on the Illumina HiSeq 2500 platform (Illumina Inc., San Diego, CA, USA) at Novogene (Beijing, China) using high-output mode with the paired-end method after library construction (NEB Next Ultra DNA Library Prep Kit). DADA2 was used to process raw sequencing reads for each sample (clean data), infer the unique amplicon variant (ASV) through error-corrected reads further quality control through the error model, and filter chimeras using the DADA2 pipeline (52). Subsequently, the sequences were filtered, trimmed, and truncated at 210 bp of forward. Then, based on the Bayesian algorithm, we used the SILVA reference database (v.12.8) to classify representative sequences from each ASV (53). Non-bacterial ASVs (chloroplast, mitochondria, unknown, Archaea, and plants) and ASVs with fewer than two reads were also removed.

## Quantitative microbial nitrogen cycling

Due to the presence of legumes in the experiment, we focused on the changes in nitrogen-related functional genes of microbes in the root zone soil (54). We selected five functional genes that represented the processes of nitrogen fixation, nitrification, and denitrification, including *nifH*, *AOA*, *AOB*, *nosZ*, *nirS*, and *nirK* (55). Quantification was carried out using the Thermo QuantStudio 6 Flex Real-Time PCR System. The primer sequences and amplification programs used for each nitrogen cycling-related gene are shown in Table S1. The reaction volume of 20 µL contained 10 µL of Maxima SYBR Green qPCR Master Mix (Thermo), 0.5 µL of forward primer (10 µmol), 0.5 µL of reverse primer (10 µmol), 10–20 ng of DNA template, and 8 µL of sterile ultrapure water. Each primer set was conducted in triplicate qPCRs at a threshold cycle (CT) of 31 as the detection limit. Finally, we averaged the three values, and each functional gene was standardized [0,1] by the min-max approach (56).

## Statistical analysis

Statistical analyses were implemented in R 4.1.2 unless otherwise stated. α-diversity was calculated for each sample using richness, Shannon, and Inverse Simpson indices based

on the normalized ASV abundance table using the rarefied method (57). Significance testing between the two groups was conducted using the Wilcoxon test, with an adjustment of the *P*-value for false discovery rate (FDR) (58). Bacterial α-diversity was evaluated by calculating the Bray-Curtis distance matrices and visualized by principal coordinate analysis (PCoA) or nonmetric multidimensional scaling analysis (NMDS). These analyses were then subjected to MRPP, one-way analysis of variance (ANOSIM), and PERMANOVA using the Adonis function of the "vegan" package in R to compare community structure under monoculture and grass-legume mixtures (59). To clarify the differences in composition between monoculture and mixed culture in classification groups, and evaluate whether certain key bacterial taxa exhibit differential abundance across different soil dilution gradients, the following analyses were conducted. First, "LEfSe" (60) was employed to compare taxa that displayed significant changes under various dilution gradients. Next, changes in bacterial abundance at the phylum and genus levels among different plant combinations were directly compared. Finally, enriched microbial taxa under different soil dilution gradients in both monoculture and mixture were visualized via a tree map using the "ggraph" package (61). To further investigate whether changes in productivity resulting from grass-legume mixtures were attributable to specific microbial taxa, we utilized the DESeq2 package (62) in R to identify ASVs that exhibited significant differences between mixtures and monoculture conditions. Co-occurrence networks of root zone bacteria in grass-legume mixtures and monocultures were constructed, with a focus on significantly different ASVs. Spearman correlation scores were computed, retaining only correlations that were both statistically significant ($P < 0.01$) and robust (Spearman's $r > 0.7$ or $r < -0.7$). Network visualization and analysis were performed using Gephi, where each node represented an ASV, and each edge indicated a strong and significant correlation between the two nodes (63). Phylogenetic trees were annotated and visualized in iTOL (Interactive Tree Of Life, an online tool to display and operation of evolutionary trees) to determine whether ASVs exhibiting significant differences are closely related to *P. stutzeri A1501* (64). Subsequently, the correlation between the identified ASVs and yield as well as nitrogen fixation genes was assessed. The Pearson correlation coefficient was calculated using the cor.test function in R (65), and the resulting correlation matrix was visualized as a heatmap using the "corrplot" package (66). To predict the geochemical material cycle processes of the bacteria, the prokaryotic taxonomic group (FAPROTAX) functional annotation database was employed (67). When assessing the potential contribution of neutral processes to microbial community assembly, we used neutral assembly (dominance test) models to predict the relationship between the occurrence frequency of ASVs and their relative abundance (68, 69). The model evaluated whether the microbial assembly process of the bacterial community conformed to a niche-based process (prediction outside the model) or neutral model (prediction inside the model).

## ACKNOWLEDGMENTS

This work was supported by the National Key Research and Development Program of China (Grant No.: 2021YFD1900500), the National Science Foundation for Excellent Young Scholars of China (Grant No.: 42122050), and the National Science Foundation of China (Grant Nos.: 42077222, 41830755, and 41807030).

All authors contributed intellectual input and assistance to this study and the manuscript preparation. S.J. developed the original framework. Y.L. performed the experiments with the help of Y.Y., Y.A., X.L., H.G., and Z.P.; Y.L. carried out the data analysis and wrote the paper with the help of S.J. and G.W.

## AUTHOR AFFILIATIONS

[1]National Key Laboratory of Crop Improvement for Stress Tolerance and Production, Shaanxi Key Laboratory of Agricultural and Environmental Microbiology, College of Life Sciences, Northwest A&F University, Yangling, Shaanxi, China

[2]Gansu Vocational College of Agriculture, Lanzhou, China

## AUTHOR ORCIDs

Yu Liu  http://orcid.org/0000-0002-9362-6306
Shuo Jiao  http://orcid.org/0000-0002-1367-6769

## FUNDING

| Funder | Grant(s) | Author(s) |
| --- | --- | --- |
| MOST \| National Key Research and Development Program of China (NKPs) | 2021YFD1900500 | Shuo Jiao |
| National Science Foundation of Excellent Young Scholars of China | 42122050 | Shuo Jiao |
| National Science Foundation of China | 42077222, 41830755, 41807030 | Shuo Jiao |

## AUTHOR CONTRIBUTIONS

Yu Liu, Data curation, Investigation, Methodology, Project administration, Writing – original draft | Wei Yan, Investigation, Methodology | Tongyao Yang, Investigation, Methodology, Project administration | Yining An, Investigation, Methodology | Xiao-meng Li, Investigation, Methodology | Hang Gao, Investigation, Methodology | Ziheng Peng, Conceptualization, Investigation, Methodology | Gehong Wei, Funding acquisition, Resources, Validation | Shuo Jiao, Conceptualization, Funding acquisition, Project administration, Resources, Supervision, Validation

## DATA AVAILABILITY

The raw sequence data reported in this paper have been deposited in the Genome Sequence Archive (70) in the National Genomics Data Center (71), China National Center for Bioinformation/Beijing Institute of Genomics, Chinese Academy of Sciences, under BioProject accession no. PRJCA018459 and are publicly accessible at https://ngdc.cncb.ac.cn/gsa.

## ADDITIONAL FILES

The following material is available online.

### Supplemental Material

**Supplemental material (mSystems00755-23-s0001.docx).** Supplemental figures and tables.

### Open Peer Review

**PEER REVIEW HISTORY (review-history.pdf).** An accounting of the reviewer comments and feedback.

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
