## [Reviewer comments · mSystems]

Grass-legume mixtures maintain forage biomass under microbial diversity loss via gathering *Pseudomonas* in root zone soil

Yu Liu, Wei Yan, Tongyao Yang, Yining An, Xiaomeng Li, Hang Gao, Ziheng Peng, Gehong Wei, and Shuo Jiao

Corresponding Author(s): Shuo Jiao, Northwest A&F University

Review Timeline:

Submission Date:	July 18, 2023
Editorial Decision:	September 13, 2023
Revision Received:	September 19, 2023
Accepted:	September 22, 2023

Editor: Nick Bouskill

Reviewer(s): The reviewers have opted to remain anonymous.

Transaction Report:

DOI: <https://doi.org/10.1128/msystems.00755-23>

September 13, 2023

Dr. Shuo Jiao
Northwest A&F University
Yangling
China

Re: mSystems00755-23 (Grass-legume mixtures maintain forage biomass under microbial diversity loss via gathering *Pseudomonas* in root zone soil)

Dear Dr. Shuo Jiao:

Thank you for submitting your manuscript to mSystems. We have completed our review and you can find the reviewer comments below. Both reviewers believe the article is of general interest to mSystem's readership, but both request revisions and additions prior to further consideration. Please pay particular attention to reviewer 1's point about additional discussion.

Also, please note that the accession numbers for the sequences archived in the Beijing Institute of Genomics (BIG) Data Center should be provided in the data accessibility section of the manuscript.

Preparing Revision Guidelines

Please return the manuscript within 60 days; if you cannot complete the modification within this time period, please contact me. If you do not wish to modify the manuscript and prefer to submit it to another journal, please notify me of your decision immediately so that the manuscript may be formally withdrawn from consideration by mSystems.

Sincerely,

Nick Bouskill

Editor, mSystems

Reviewer comments:

Reviewer #1 (Comments for the Author):

In the manuscript titled "Grass-legume mixtures maintain forage biomass under microbial diversity loss via gathering *Pseudomonas* in root zone soil" authored by Liu and colleagues, the researchers investigated how the legume-grass mixture assist neighboring grass to resist the biomass decline when soil microbial diversity is lost. The experiment was conducted by constructing the soil biodiversity gradient and the full factorial combination of legumes and two grasses. The authors discovered an interesting phenomenon that when grass was monocultured, biomass was lost as soil biodiversity declined, but biomass remained stable when grass was mixed with legume. This research enhances our comprehension of how aboveground biomass is maintained in the face of disturbance and provides a deeping understanding for the role of legumes.

The author makes a lot of discussion on microbe especially *Pseudomonas*, but ignores the plant host in the Discussion. Just as when two grasses are mixed with legumes, only one grass (*Elymus dahuricus* Turcz.) shows a stable biomass when microbial diversity is lost. Does this mean that it is affected by plant identity? Here authors need to further add the possible differences between the two grasses in the Discussion and also show the results of *Festuca elata* Keng. in the Results.

I have the following minor suggestions.

Line 25: What specific combinations of legumes and grasses were made? Such as full-factorial combinations of legume and two grasses?

Line 27: legume and which grass of two grasses? Be specific.

Line 31: add (nifH) between nitrogen-fixing and gene.

Line 37: remove in the microbial community.

Line 54: add reference.

Line 77-80: add reference.

Line 109: Replace ',' with '.'

Line 127: Replace 'under different diversity conditions' with 'across soil dilution gradients'

Line 365: Why is 10⁻² used here and 10² in the figure?

Line 378: add spaces between numbers and units

Line 388: RH mean?

Line 370: 5 plots mean 5 replicates? Does it contradict 4 replicates in Line 402?

Line 421: As far as I know, using data2 to process amplicon data does not need to consider 97% similarity. A deeper understanding is needed about data2 processes.

Line 438-482: Active tense is used too much. Except for necessary explanations, all other sentences are changed to the passive tense.

Line 682: Replace ',' with '.'

Line 683: Which method was used to test for significant differences? It also needs to be shown in the figure legend.

Fig. 1ab: Do 'M' in (a) and 'M_mono' in (b) are the same meaning? If yes, keep the same abbreviation. There is no unit in Biomass.

Fig. 1c: Soil dilution gradient needs be explained in figures legend. What does 100, 102, 104, 106 mean?

Fig. 2: Before showing the ab sub-legend, one sentence as the title of Fig. 2 need be illustrated.

Line 694: enriched in monoculture and/or in mixture?

Line 695: in the corresponding soil dilution gradient?

Line 696: The plot called tree plot?

Fig. 2a: The indicated taxa are Class level?

Fig. 3b: There are no unit in nirH abundance and ASV abundance.

Fig. 4a: The color of the negative interaction cannot be seen clearly, so change it to another color.

Fig. 4b: The parameter of network topology do not need to be abbreviated, such as positive interaction and edge number.

Fig. 6: Remove underline in microbial functions.

Reviewer #2 (Comments for the Author):

The manuscript by Liu et al. aims to examine whether grass-legume mixtures play a positive role in increasing plant biomass under low soil microbial diversity. The authors created a gradient of soil microbial diversity using a dilution method, and then utilized various combinations of legume and grass to investigate the role of legumes in influencing the biomass of neighboring plants. They found that grass-legume mixtures enhanced the biomass of grass under low soil microbial diversity. They revealed

a significant enrichment of *Pseudomonas* (ASV53) in the grass-legume mixtures, which was positively associated with plant biomass and nitrogen-fixing gene abundance. Further functional predictions indicated that grass-legume mixtures may increase the abundance of N-related and phototrophy-related microbial communities in the root zone soil. Investigating the effects of plant-plant and plant-microbiomes interactions on plant performance under different microbial diversity and their associated ecological mechanisms represent a very interesting and promising research topic. Results from this study can offer an important knowledge for engineering microbial functions to improve crop production in a sustainable way. The results of this article are highly interesting, and the authors have employed appropriate bioinformatic and statistical analyses. I just have a few minor comments that might be worth thinking before the publication. Specific comments are provided below:

L36-37: I would recommend reconsidering the evidences supporting conclusion on human pathogens, as this isn't the paper's main focus and functional prediction and genus-level identification of pathogens based on 16S is limited in reliability.

L196-198: It would be great if you could provide the similarity between these two sequences (i.e. ASV53 and A1501). I would suggest enlarging the text in Fig. 3c as it looks not very clear.

L360: "soil microbial communities" to "the diversity of soil microbial communities"

L407-409: Please add more detailed information on the PCR amplification.

L496-498: Please add the accession number for raw sequence data.

Response and actions taken with respect to the reviewer comments for:

mSystems00755-23

Title: Grass-legume mixtures maintain forage biomass under microbial diversity loss
via gathering *Pseudomonas* in root zone soil

Authors: Yu Liu¹, Wei Yan¹, Tongyao Yang, Yining An, Xiaomeng Li, Hang Gao,
Ziheng Peng, Gehong Wei^{*}, Shuo Jiao^{*}

Reviewer #1:

1. This research enhances our comprehension of how aboveground biomass is maintained in the face of disturbance and provides a deeping understanding for the role of legumes.

Response: Thank you very much for the positive comments.

2. The author makes a lot of discussion on microbe especially *Pseudomonas*, but ignores the plant host in the Discussion. Just as when two grasses are mixed with legumes, only one grass (*Elymus dahuricus* Turcz.) shows a stable biomass when microbial diversity is lost. Does this mean that it is affected by plant identity? Here authors need to further add the possible differences between the two grasses in the Discussion and also show the results of *Festuca elata* Keng. in the Results.

Response: Thank you very much for providing valuable feedback. As you pointed out, the role of plant hosts is indeed crucial in our experiment. I have incorporated your suggestions into the revised version, addressing why there was no similar increase in biomass for *Festuca elata* Keng. compared to *Elymus dahuricus* Turcz. Furthermore,

you mentioned the absence of *Festuca elata* Keng. in the subsequent results analysis. This was due to our observation during biomass analysis that the *Medicago sativa* L. and *Elymus dahuricus* Turcz. mixture increased the biomass of *Elymus dahuricus* Turcz., while the biomass of *Festuca elata* Keng. did not show an increase when mixed with *Medicago sativa* L. or grown in monoculture. Therefore, we did not investigate the underlying mechanisms for *Festuca elata* Keng. in this context.

I added the relevant content in lines 278-296, which are shown below: We first compared the variations in biomass across different plant combinations. We observed that the M and P mixture increased the biomass of P, while mixtures had no significant effect on the biomass of G. Similar observations were made with plant combinations GP and MGP. This is the reason why we focused our subsequent analysis exclusively on the individual growth of M and P and their mixed cultivation as MP. The occurrence of these patterns is not only attributed to the crucial role of the soil microbial community, which we have been investigating but also to the inherent characteristics of the plants themselves (1). As commonly used grass species in establishing artificial grasslands, G and P exhibit preferences for specific soil environments and tolerances. Empirical knowledge from livestock farming indicates that P thrives in nutrient-rich soils (2). When P is mixed with M, the nitrogen-fixing capacity of M provides a more abundant nitrogen source, creating a nutrient-rich environment that favors the growth of P. On the contrary, G responds sensitively to soil fertilization, and excessively high nitrogen content in the soil can be detrimental to its growth (3). Thus, a mixture with M does not increase G biomass. In addition, P

demonstrates strong adaptability to temperature, while G prefers cooler environments

(4). The controlled temperature conditions in our study may not have reached the optimal growth temperature for G. Consequently, there were no significant differences in biomass, whether in monoculture or mixture.

3. What specific combinations of legumes and grasses were made? Such as full-factorial combinations of legume and two grasses? (Line 25).

Response: Yes, our plant combinations included one legume, *Medicago sativa* L., and two grasses, *Elymus dahuricus* Turcz. and *Festuca elata* Keng.. These plants were grown individually and in pairwise combinations, as well as all three together, making a total of seven plant combinations. In the revised version, I will incorporate the full-factorial interactions as provided here to elucidate the experimental design.

Line 23-25 “To address this, we employed a dilution method to create a gradient of decreasing microbial diversity in soil, and utilized full-factorial combinations of legume and two grasses to investigate the crucial role of legumes in the mixture.”

4. legume and which grass of two grasses? Be specific. (Line 27).

Response: Done. I have clarified the plant combinations that exhibit biomass enhancement to avoid any confusion.

Line 26-27 “The results showed that compared to monoculture, the mixture of *Medicago sativa* L. and *Elymus dahuricus* Turcz. enhanced the biomass of grass species under conditions of soil microbial diversity loss.”

5. add (nifH) between nitrogen-fixing and gene. (Line 31).

Response: Done.

6. remove in the microbial community. (Line 37).

Response: Done.

7. add reference. (Line 54).

Response: Done.

8. add reference. (Line 77-80).

Response: Done.

9. Replace ',' with '!'. (Line 109).

Response: Done.

10. Replace 'under different diversity conditions' with 'across soil dilution gradients'.

(Line 127).

Response: Done.

11. Why is 10^{-2} used here and 10^2 in the figure? (Line 365).

Response: I appreciate your valuable feedback. I used " 10^{-2} " here to describe the final concentration after soil suspension dilution, whereas the " 10^2 " in the figure represents

the dilution factor applied to the soil suspension. To prevent any potential confusion, I have made a consistent modification to describe it in terms of the dilution factor.

12. add spaces between numbers and units. (Line 378).

Response: Done.

13. RH mean? (Line 388).

Response: I apologize for the oversight. Here, "RH" refers to Relative Humidity, and I have added detailed explanations in the revised version for clarity.

14. 5 plots mean 5 replicates? Does it contradict 4 replicates in Line 402? (Line 370).

Response: Yes, the reference to "5 plots" represents 5 replicates. During the experiment, each plant combination under each dilution gradient had 5 replicates. However, based on the growth conditions of the plants and considering the consistency of sample sizes across all groups, we ultimately selected 4 replicates for further analysis.

15. As far as I know, using dada2 to process amplicon data does not need to consider 97% similarity. A deeper understanding is needed about dada2 processes. (Line 421).

Response: Thank you for your valuable feedback. In the revised version, I have rewritten the procedure for obtaining the ASV table.

Line 439-446 "DADA2 was used to process raw sequencing reads for each sample

(clean data), infer the unique amplicon variant (ASV) through error-corrected reads further quality control through the error model, and filter chimeras using the DADA2 pipeline (5). Subsequently, the sequences were filtered, trimmed, and truncated at 210 bp of forward. Then, based on the Bayesian algorithm, we used the SILVA reference database (v.12.8) to classify representative sequences from each ASV (6). Non-bacterial ASVs (chloroplast, mitochondria, unknown, Archaea and plants) and ASVs with fewer than two reads were also removed.”

16. Active tense is used too much. Except for necessary explanations, all other sentences are changed to the passive tense. (Line 438-482).

Response: Done.

Line 484-486 “Co-occurrence networks of root zone bacteria in grass-legume mixtures and monocultures were constructed, with a focus on significantly different ASVs.”

Line 488- 490 “Network visualization and analysis were performed using Gephi, where each node represented an ASV, and each edge indicated a strong and significant correlation between the two nodes”

17. Replace ',' with '!'. (Line 682).

Response: Done.

18. Which method was used to test for significant differences? It also needs to be

shown in the figure legend. (Line 683).

Response: Done. Analyses of Variances (ANOVAs) were conducted to test for significant differences.

19. Do 'M' in (a) and 'M_mono' in (b) are the same meaning? If yes, keep the same abbreviation. There is no unit in Biomass. (Fig. 1ab).

Response: Yes, the 'M' in (a) and 'M_mono' in (b) have the same meaning, I have renamed them in the revised version for clarity, and the unit of Biomass was added.

20. Soil dilution gradient needs be explained in figures legend. What does 100, 102, 104, 106 mean? (Fig. 1c).

Response: Done. The values indicated in the figure, namely 10^0 , 10^2 , 10^4 , and 10^6 , represent the dilution gradients of the soil suspension. Specifically, 10^0 signifies the undiluted original mother solution, 10^2 corresponds to a 100-fold dilution, 10^4 denotes a 10,000-fold dilution, and 10^6 signifies a 100,000-fold dilution. We have added a detailed description of the soil dilution gradient in the revised version, and the above contents are added to the legend in Figure 1.

21. Before showing the ab sub-legend, one sentence as the title of Fig. 2 need be illustrated. (Fig. 2).

Response: Done.

22. enriched in monoculture and/or in mixture? (Line 694).

Response: I apologize for any misunderstanding, it should be enriched in monoculture and mixtures.

23. in the corresponding soil dilution gradient? (Line 695).

Response: Yes, I will correct it in the revised version to prevent misunderstandings.

24. The plot called tree plot? (Line 696).

Response: Yes, this is a circular packing chart, which is a variant of a tree map. To avoid any misunderstandings, I will change the name to the former in the revised version.

25. The indicated taxa are Class level? (Fig. 2a).

Response: No, what is being presented here are microbial taxa enriched in both monoculture and mixtures, ranging from the genus level to the class level. We have added the above contents to the legend in Figure 2, in the revised version.

26. There are no unit in nifH abundance and ASV abundance. (Fig. 3b).

Response: Done.

27. The color of the negative interaction cannot be seen clearly, so change it to another color. (Fig. 4a).

Response: Done.

28. The parameter of network topology do not need to be abbreviated, such as positive interaction and edge number. (Fig. 4b).

Response: Done.

29. Remove underline in microbial functions. (Fig. 6).

Response: Done.

Reviewer #2:

1. The results of this article are highly interesting, and the authors have employed appropriate bioinformatic and statistical analyses. I just have a few minor comments that might be worth thinking before the publication.

Response: Thank you very much for the positive comments.

2. I would recommend reconsidering the evidences supporting conclusion on human pathogens, as this isn't the paper's main focus and functional prediction and genus-level identification of pathogens based on 16S is limited in reliability. (Line 36-37).

Response: Thank you for your valuable comments. Predicting the abundance of human pathogenic microorganisms based on functional predictions may not always be accurate. I have removed this portion from the abstract as per your suggestion in the

revised version.

3. It would be great if you could provide the similarity between these two sequences (i.e. ASV53 and A1501). I would suggest enlarging the text in Fig. 3c as it looks not very clear. (Line 196-198).

Response: Done. As the sequence information for ASV53 is V4-V5 region derived from high throughput 16S rRNA sequencing data, I selected a segment of the 16S rRNA from the A1501 whole genome sequence and aligned it with ASV53. The results are as follows. The similarity between ASV53 and A1501 was 98.12%. The text in Fig. 3c was enlarged in the revised version.

```
A1051 520 TACGAAAGGTTGCAAGCGTTAATCGGAATTACTGGGCGTAAAGCGCGGTAGGTGGTTTCGTTAAGTTGGATGTGAAAAGCCCGGGCTCAACCTGGGAACCTGCATCCAAAACCTGGCGAGCTA 647
ASV53 1 TACGAAAGGTTGCAAGCGTTAATCGGAATTACTGGGCGTAAAGCGCGGTAGGTGGTTTCGTTAAGTTGGATGTGAAAAGCCCGGGCTCAACCTGGGAACCTGCATCCAAAACCTGGCGAGCTA 120

A1051 640 GAATATGGCAGAGGGTGGTGGAAATTCCTGTGTAGCGGTGAAATGCGTAGATATAAGAAAGGAACACCAAGTGGCGAAAGCGACCACCTGGGCTAATACTGACACTGAGGTGCAAAAGCGTG 767
ASV53 121 GAATATGGTAGAGGGTGGTGGAAATTCCTGTGTAGCGGTGAAATGCGTAGATATAAGAAAGGAACACCAAGTGGCGAAAGCGACCACCTGGGCTAATACTGACACTGAGGTGCAAAAGCGTG 240

A1051 768 GGGAGCAAAACAGGATTAGATACCCCTGGTAGTCCACGCCGTAACGATGTCGACTAGCCGTTGGGATCCTTGAAGATCTTAGTGGCGCAGCTAACGCATTAAGTCGACCCTGGGGAGTAC 887
ASV53 241 GGGAGCAAAACAGGATTAGATACCCCTGGTAGTCCACGCCGTAACGATGTCGACTAGCCGTTGGGAGCCCTTGAAGCTCTTAGTGGCGCAGCTAACGCATTAAGTTGACCCTGGGGAGTAC 360

A1051 888 GGCCGCAAGGTTA 900
ASV53 361 GGCCGCAAGGTTA 373
```

4. "soil microbial communities" to "the diversity of soil microbial communities".
(Line 360).

Response: Done. Modify the sentence to “The gradient of the diversity of soil microbial communities was constructed using the dilution gradient method of bacterial suspension.”

5. Please add more detailed information on the PCR amplification. (Line 407-409).

Response: Done. We performed 16S rDNA amplicon sequencing of the root zone soil

samples. For prokaryotic amplicon preparation, the V4-V5 region of the 16S rDNA gene was amplified by PCR using 515F (5'-GTGCCAGCMGCCGCGGTAA-3') and 907R (5'-CCGTCAATTCCTTTGAGTTT-3') primers. PCR reactions were performed in triplicate in a 15 μ L reaction mixture which contained 7.5 μ L of Phusion® High-Fidelity PCR Master Mix (New England Biolabs, Ipswich, MA, USA), 1 μ L of forward and reverse primers (3 mM), 2.5 μ L of template DNA (5 ng μ L⁻¹), and 4 μ L of ddH₂O. PCR conditions were: 1 cycle \times 98 °C for 1 min; 30 cycles \times 98 °C for 10 s, 50 °C for 30 s, and 72 °C for 30 s; 1 cycle \times 72 °C for 5 min. The quality of amplicons was detected through electrophoresis on 2 % agarose gel, and the purity of amplicons was ensured using Qiagen Gel Extraction Kit (Qiagen, Hilden, Germany). The triplicate PCR reactions for each sample were combined and quantified on a QuantiFluor™-ST Fluorometer (Promega, Madison, WI, USA) following the manufacturer's protocol. 16S rDNA sequencing was performed on the Illumina HiSeq® 2500 platform (Illumina Inc., San Diego, CA, USA) at Novogene (Beijing, China) using high-output mode with the paired-end method after library construction (NEB Next® Ultra DNA Library Prep Kit).

6. Please add the accession number for raw sequence data. (Line 496-498).

Response: Done. The raw sequence data reported in this paper have been deposited in the Genome Sequence Archive (70) in the National Genomics Data Center (71), China National Center for Bioinformation / Beijing Institute of Genomics, Chinese Academy of Sciences, under BioProject accession no. PRJCA018459 and are publicly

accessible at <https://ngdc.cncb.ac.cn/gsa>.

Reference

1. Ren C, Zhang W, Zhong Z, Han X, Yang G, Feng Y, Ren G. 2018. Differential responses of soil microbial biomass, diversity, and compositions to altitudinal gradients depend on plant and soil characteristics. *Science of The Total Environment* 610-611:750-758.
2. Wu W-D, Liu W-H, Sun M, Zhou J-Q, Liu W, Zhang C-L, Zhang X-Q, Peng Y, Huang L-K, Ma X. 2019. Genetic diversity and structure of *Elymus tangutorum* accessions from western China as unraveled by AFLP markers. *Hereditas* 156:8.
3. Zhang Y, Yang J-y, Wu H-l, Shi C-q, Zhang C-l, Li D-x, Feng M-m. 2014. Dynamic changes in soil and vegetation during varying ecological-recovery conditions of abandoned mines in Beijing. *Ecological Engineering* 73:676-683.
4. Li L, Li X, Lin L, Wang Y, Xue WJCJoPE. 2011. Comparison of chlorophyll content and fluorescence parameters of six pasture species in two habitats in China. *Chinese Journal of Plant Ecology* 35:672-680.
5. Wen T, Zhao M, Yuan J, Kowalchuk GA, Shen Q. 2021. Root exudates mediate plant defense against foliar pathogens by recruiting beneficial microbes. *Soil Ecology Letters* 3:42-51.
6. Jiao S, Chu H, Zhang B, Wei X, Chen W, Wei G. 2022. Linking soil fungi to bacterial community assembly in arid ecosystems. *iMeta* 1:e2.

September 22, 2023

Dr. Shuo Jiao
Northwest A&F University
Yangling
China

Re: mSystems00755-23R1 (Grass-legume mixtures maintain forage biomass under microbial diversity loss via gathering *Pseudomonas* in root zone soil)

Dear Dr. Shuo Jiao:

Your manuscript has been accepted, and I am forwarding it to the ASM Journals Department for publication. For your reference, ASM Journals' address is given below. Before it can be scheduled for publication, your manuscript will be checked by the mSystems production staff to make sure that all elements meet the technical requirements for publication. They will contact you if anything needs to be revised before copyediting and production can begin. Otherwise, you will be notified when your proofs are ready to be viewed.

If you would like to submit a potential Featured Image, please email a file and a short legend to msystems@asmusa.org. Please note that we can only consider images that (i) the authors created or own and (ii) have not been previously published. By submitting, you agree that the image can be used under the same terms as the published article. File requirements: square dimensions (4" x 4"), 300 dpi resolution, RGB colorspace, TIF file format.

We recognize that the video files can become quite large, and so to avoid quality loss ASM suggests sending the video file via <https://www.wetransfer.com/>. When you have a final version of the video and the still ready to share, please send it to mSystems staff at msystems@asmusa.org.

Sincerely,

Nick Bouskill
Editor, mSystems

Journals Department
E-mail: mSystems@asmusa.org